# Circulating Tumor Cells in Early and Advanced Breast Cancer; Biology and Prognostic Value

**DOI:** 10.3390/ijms21051671

**Published:** 2020-02-29

**Authors:** Anna Fabisiewicz, Malgorzata Szostakowska-Rodzos, Anna J. Zaczek, Ewa A. Grzybowska

**Affiliations:** 1Department of Molecular and Translational Oncology, Maria Sklodowska-Curie National Research Institute of Oncology, Roentgena 5, 02-781 Warsaw, Poland; anna.fabisiewicz@coi.pl (A.F.); malgorzata.szostakowska@coi.pl (M.S.-R.); 2Laboratory of Translational Oncology, Intercollegiate Faculty of Biotechnology, Medical University of Gdańsk, Gdańsk, Debinki 1, 80-211 Gdansk, Poland; azaczek@gumed.edu.pl

**Keywords:** CTC, breast cancer metastasis, prognostic value

## Abstract

Breast cancer metastasis is the leading cause of cancer deaths in women and is difficult to combat due to the long periods in which disseminated cells retain a potential to be re-activated and start the relapse. Assessing the number and molecular profile of circulating tumor cells (CTCs) in breast cancer patients, especially in early breast cancer, should help in identifying the possibility of relapse in time for therapeutic intervention to prevent or delay recurrence. While metastatic breast cancer is considered incurable, molecular analysis of CTCs still have a potential to define particular susceptibilities of the cells representing the current tumor burden, which may differ considerably from the cells of the primary tumor, and offer more tailored therapy to the patients. In this review we inspect the routes to metastasis and how they can be linked to specific features of CTCs, how CTC analysis may be used in therapy, and what is the current status of the research and efforts to include CTC analysis in clinical practice.

## 1. Introduction

Breast cancer (BC) has the highest incidence rate and is the leading cause of cancer death in women, however, the overall mortality is not as high as in the other types of cancer [1]. This is because estrogen-positive (luminal) cancers, which constitute about 75% of all cases are susceptible to hormonal therapy, and have relatively low metastatic potential. In general, different biological subtypes of breast cancer display differences in propensity to metastatic spread. Biological subtypes are classified according to the expression of steroid receptors (estrogen and progesterone) and HER-2 (human epidermal growth factor receptor 2). In luminal cancers metastasis is not frequent, but this group is heterogeneous, comprising luminal A, B and C, with different aggressiveness and treatment options [2]. There is a peak of relapses at about 2–3 years after the surgery, but the rate of metastasis remains relatively high after the peak, because disseminated cells may stay dormant even for as long as 20 years and start to proliferate upon some triggering event [3]. In tumors with amplified HER2 (about 20–30% of cases) prognosis is relatively worse than in luminal BC, although they can be treated with targeted anti-HER2 therapy. Triple-negative breast cancer (TNBC, ER- PR- HER2-), which constitutes about 15% of all cases, is heterogeneous itself, comprising different subtypes, including basal like, immune-modulatory, luminal androgen receptor, mesenchymal and claudin low. Despite these differences, it is always highly aggressive, associated with chemotherapeutic resistance and low survival rates. It tends to disseminate quickly (also with a peak in the first 2–3 years after surgery) and the percentage of metastases is high [4,5]. However, in contrast to luminal cancers, the rate of distant metastases in TNBC becomes significantly lower after five years, thus dormancy of disseminated tumor cells (DTCs) seems to be less common in this subtype.

These differences raise the question what are the routes to metastasis in breast cancer, what can we learn about them by analyzing circulating tumor cells (CTCs) and how the knowledge gained by CTC analysis may impact therapeutic decisions.

## 2. Routes to Metastasis in Breast Cancer (BC)

There are two main hypotheses describing cancer dissemination: linear progression, according to which, metastatic potential is gained gradually, and the whole process proceeds in steps and needs time, and parallel progression, according to which dissemination takes place early on, even before clinical manifestation of the tumor [6,7] (Figure 1). In the linear progression model, the evolving primary tumor gives rise to metastases due to increasingly aggressive and invasive phenotypes of the subclones of tumor cells. This model is in agreement with the postulated role of the epithelial-to-mesenchymal transition (EMT) in cancer and assumes active degradation of the stroma and migration into blood vessels by the motile, invasive single cells that broke loose from the epithelial monolayer. In contrast to that, parallel progression postulates intravasation by passive shedding, which may take place shortly after the angiogenic switch, occurring during the early, pre-invasive stage of tumor development [8,9].

Although the concept of linear progression is well established, recently many reports support parallel progression hypothesis [10]. Different timelines and dynamics of breast cancer metastasis in different subtypes might suggest differences in the mode of dissemination, but more convincing evidence is needed to prove that it is the case. At the moment it seems that parallel progression is better described for HER2-positive tumors [11,12] but also occurs in ER-positive type [13]. Detailed genomic analyses of the samples from 6 ER-positive breast cancer patients revealed not only the co-occurrence of linear and parallel progression, but also provided evidence for metastasis-to metastasis seeding [13].

TNBC cancers were always associated with EMT as a causative agent for dissemination, hence linear progression is the most probable in this case [14]. However, (as discussed below) the role of EMT in cancer metastasis is much debated and not as straightforward as assumed [15,16].

Accepting the parallel progression model, even if only for some subtypes or specific cases of breast cancer, calls for a close scrutiny of the current diagnostic and therapeutic procedures and has implications for the CTC research. Early dissemination, occurring before the appearance of any symptoms, precludes prevention, unless we consider precautionary screening of the healthy population, but even in this case, the probability of detecting CTCs in time is negligible. The group of Terstappen [17] presented mathematical model of breast cancer progression, demonstrating that to detect cancer before distant metastasis the sensitivity of CTC detection should be improved at least 15-fold, and the corresponding lesion giving rise to metastatic CTCs can be only 2.7 mm in diameter or smaller. However, even if CTC data could not be used for the prevention of metastasis, they still carry important information, with diagnostic and therapeutic relevance. CTCs represent tumor cells’ population currently present in the circulation and, as such, can serve as a source of invaluable information, considering tumor properties, more accurate, recent and relevant than tissue analysis from the primary tumor obtained during surgery. Learning from that input, we can modify the treatment and either hold the potential metastases in check, by knowing their biology, or, in more advanced cases, tailor the treatment according to the specific properties and sensitivities of the circulating cells. This can be especially important in case of ER-positive breast cancer, which tends to lose responsiveness to estradiol. CTC screening may detect this change and flag the necessity for therapy modification. Gaining more knowledge on the issue if some routes of metastasis are more likely in specific subtypes of breast cancer would be invaluable for the CTC data interpretation.

### 2.1. Factors Affecting the Number of Circulating Tumor Cells (CTCs) in the Blood

The sheer number of CTCs circulating in the blood of a patient is the simplest and, as shown below, statistically significant factor in assessing the outcome. In breast cancer, CTCs are detected in about 20–30% of early and around 60% of advanced patients. Interestingly, breast cancer subtypes were found to have no impact on absolute CTC numbers as well as CTC positivity rates, except for the very high CTC counts observed more frequently in luminal A and TNBC and the lower prognostic impact for the HER2-positive disease [18]. However, we still need more information enabling us to compare and assess CTC numbers reliably. CTC numbers can vary in the blood of the same patient and we need to determine factors influencing it. We have to consider simple methodology factors like blood drawing (arterial vs. venous, the time of day, the technique) [19] or clinical factors, including overall patients’ status and clinical features but also the important issue of CTCs mobilization by therapeutic intervention, like radiotherapy, surgery or even biopsy [20,21]. There is also a trivial, but disturbing issue of the proper categorization of CTCs; considering the multitude of techniques and relatively subjective methods of CTCs’ assessment, we probably should consider some ways of uniformization of the CTC count.

As reviewed below, the enumeration of CTCs represents an effective prognostic and predictive biomarker. To what exactly should we attribute this prognostic power? To some extent, probably, it can be explained by the metastatic potential of these cells, but almost certainly it is not always the case. Alternatively, CTC shedding may be associated with the increased aggressiveness of the tumor. It would be beneficial to accumulate more knowledge in regard to the shedding process; does it occur constantly or in waves and what stimuli trigger it? For example, it was recently demonstrated that CTCs’ presence correlates with low red blood cell count and that the treatment with the monoclonal antibody denosumab is associated with the absence of CTCs [22].

Clarification of the factors determining the CTC release pattern is thus an important goal that can be achieved only by systematic longitudinal studies of CTC numbers in a large patients’ cohorts.

### 2.2. Genotype and Phenotype of CTCs in BC

Besides enumeration, new techniques, which enable to efficiently isolate CTCs from patients’ blood provide the possibility of a detailed molecular analysis, including the detection of genetic and epigenetic changes, expression profiling and phenotype screening. Fortunately, due to a recent development of new techniques for single-cell analysis, CTCs heterogeneity can be efficiently analyzed at all these levels.

#### 2.2.1. Genotype

Genotype analysis of CTCs in breast cancer is tuned to detect the most common driver mutations and to assess the difference between CTCs and primary tumor. Frequently mutated genes in breast cancer CTCs include *TP53* (mostly in TNBC), *PIK3CA*, *ERBB2*, *KRAS* and *ESR1.* Moreover, mutations in these genes display high degree of intra-patient heterogeneity [23,24]. Single cell analysis enables heterogeneity to be detected, which can be lost in bulk analysis, as described by Pestrin et al. [25], for *PIK3CA*. Thus, it is important to recognize mutations at the single-cell level, since they may give rise to metastatic clones, while remaining undetectable in bulk analysis. Similar conclusions can be drawn from the study by De Luca et al. [26]. The authors analyzed heterogeneity of isolated CTCs by whole-genome amplification and next-generation sequencing (NGS) in 50 cancer-related genes, finding substantial discordance in mutational status between CTCs and primary tumor and high intra-patient heterogeneity, with the highest number of somatic mutations for *TP53*.

Genetic changes can be detected in isolated CTCs or in cell lines isolated from CTCs. For example, genomic analysis of the CTC cell lines cultured from six patients with estrogen receptor positive BC revealed preexisting mutations in the *PIK3CA* gene and newly acquired mutations in *ESR1, PIK3CA* and *FGFR2* (fibroblast growth factor receptor gene) [27].

Interestingly, newly acquired mutations in *ESR1* are quite common, despite the fact that the primary tumor is usually ER-positive. Accordingly, the analysis of the receptor status in CTCs from metastatic patients demonstrated that most of CTCs were estrogen and progesterone receptor negative, while primary tumors were receptor-positive [28]. This probably reflects a mechanism of escape from endocrine therapy. In contrast to that, in receptor-negative BC changes in steroid receptor status between primary tumor and CTCs are usually minimal. Discordance in expression between primary tumor, CTC and metastases was also observed for HER-2 [29].

#### 2.2.2. Changes in Expression; Reversible Re-Programing

When a tumor cell enters the circulation and becomes CTC it is exposed to various stresses associated with this new microenvironment, including anoikis (apoptosis driven by the lack of adhesion), hemodynamic shear forces, oxidative stress and immune attack. Most of these cells die in the first hours [30] and only a few resistant cells are capable of extravasation and dissemination. In mouse models, roughly 2.5% of CTCs formed micro-metastases, most disappeared over time, and 0.01% of CTCs progressed to form macro-metastases [31]. Thus, the cell reversibly re-programs itself to cope with new challenges. Changes in expression include metabolic re-programing to withstand oxidative stress, cytoskeletal re-organization to withstand deformations, EMT, with a multitude of functions (to combat anoikis, confer more stemness, resistance to stress, including chemoresistance), and the expression of specific molecules to mask against immune attack (PD-L1 expression, [32]). Alternatively, some protection from these factors can be achieved by cells’ clustering.

Metabolic shift towards glycolysis is a hallmark of cancer cells. Brisotto et al., [33] tested prognostic value of metabolically altered CTCs in metastatic breast cancer (MBC) and reported some discriminating power of this test (extracellular acidification assay), although on a small number of patients. On the other hand, LeBleu et.al., [34] compared expression profiles of CTCs and primary tumor cells using a mouse model of breast cancer metastasis and reported the opposite phenomenon; a shift towards oxidative phosphorylation (OXPHOS). The OXPHOS gene set was the most differentially regulated, followed by actin cytoskeleton signaling and pyrimidine and purine metabolism pathways, while all other metabolic pathways were only minimally changed. Interestingly (and logically), the cell with a high migratory potential not only exhibits diminished proliferation status, but also usually prefers oxidative phosphorylation above glycolysis, which contradicts properties usually associated with tumors, but suits very well the purpose of dissemination from the site with limited resources by temporary re-programming which enables it to withstand harsh conditions during migration.

CTCs’ expression prolife display intra-patient heterogeneity, which suggest the existence of some distinct population of metastasis-initiating cells; indeed, the work of Baccelli et al. [35] demonstrated, using a mouse xenograft assay, that such a population in luminal breast cancer CTCs is enriched with EPCAM, CD44, CD47 and MET.

Several studies indicated that expression profiling of CTCs may have clinical applications. Prognostic significance was demonstrated for the expression of TWIST1, ALDH1, CD44, CD24 in CTCs from early breast cancer [36]. Reijm et al., developed an 8-gene expression profile in CTCs which discriminates good and poor outcome to first-line aromatase inhibitors in MBC patients [37]. Bredemeier et al., also reported gene expression signatures in CTCs that can be used to monitor response to therapy in MBC [38]. It was also demonstrated that changed gene expression during first-line systemic therapy for MBC could be a possible explanation for treatment resistance [39].

CTC expression profiling often includes epithelial and mesenchymal markers, like adhesion molecules (EpCAM, MUC1) or EMT-promoting transcription factors (TWIST1). Epithelial cell adhesion molecule (EpCAM) is used for the isolation of CTCs by immunomagnetic enrichment in the CellSearch system. This approach was shown to result in a loss of a substantial percentage of EpCAMl^ow^ CTCs, but these cells were shown to have no impact on overall survival (OS) in metastatic breast cancer. Conversely, the presence of EpCAM^high^ cells was demonstrated to be strongly related to poor survival [40]. Examples of EpCAM heterogeneous expression among CTCs from patients with MBC are presented in Figure 2. Another epithelial adhesion molecule, E-cadherin, was also shown to be associated with higher metastatic potential [41]. These findings seem counter-intuitive, since the presence of these adhesion molecules, considered the hallmarks of epitheliality, were usually interpreted as associated with non-metastatic phenotype. To explain these results, one should take a closer look at EMT and its role in cancer.

### 2.3. Epithelial-to-Mesenchymal Transition (EMT)

The epithelial-to-mesenchymal transition (EMT) is a developmental program that enables stationary epithelial cells to migrate and invade as single cells [42]. It is a multistage process associated with dynamic changes in morphology, cytoskeleton and adhesion, including the loss of apicobasal polarity and delocalization of tight junction and adherens junction proteins, such as E-cadherin and ZO-1 [43,44]. Cells adopt a spindle-shaped, mesenchymal-like morphology with upregulated expression of mesenchymal markers, such as N-cadherin, fibronectin and vimentin [45]. The molecular changes during EMT are well described in the current literature (reviewed: [46,47]). These changes are controlled by several signaling pathways (TGF-β1, Wnt, Notch, Hedgehog) and transcription factors (TWIST1, SNAIL1, SLUG, ZEB1 and/or FOXC1/2) [46,48]. 

EMT is necessary for embryonal development and wound-healing, but may be hijacked by cancer cells. Cells after EMT are more motile, invasive and resistant to stress associated with circulation, but to seed in another organ they need to regain their epithelial characteristics and undergo mesenchymal-to-epithelial transition (MET), probably to restore proliferative potential, which is diminished in cells after EMT. As mentioned earlier, the EMT/MET sequence is considered a classic metastatic route. However, there is a controversy surrounding the role of EMT in metastasis. The most recent research highlighted that EMT may not be essential for metastasis in tumors of epithelial origin. The in vivo studies on EMT role in lung metastases formation in breast cancer demonstrated that those lesions are derived from non-EMT cells, which contradicts the original EMT/MET hypothesis. Moreover, these studies also demonstrated that overexpression of miR-200 (EMT suppressor) have no impact on metastasis formation [15]. Similar research on various types of cancer lead to the conclusion that tumor cells are not definitely dependent on upregulation of EMT markers to become metastatic [49,50]. 

### 2.4. Hybrid Epithelial-Mesenchymal (E/M) Phenotype 

Recent studies categorize EMT in cancer more as a spectrum of phenotypes between mesenchymal and epithelial than a directional, complete process [51,52]. In fact, epithelial-mesenchymal plasticity (EMP), encompassing both, EMT and MET is a better term in a context of tumor cells [53]. EMP in carcinoma enables the acquisition of certain mesenchymal properties without losing all epithelial traits [54]. Such cells expressing both epithelial and mesenchymal markers are identified as hybrid epithelial-mesenchymal (hybrid E/M) after partial EMT. How is this hybrid phenotype achieved? The EMT process comprises two mutually inhibitory loops: MiR-34/SNAIL and MiR-200/ZEB. There are two computational models for role of those loops in EMT and partial EMT. First model proposes that both loops function as binary switches that initiate (MiR-34/SNAIL) and stabilize (MiR-200/ZEB) EMT. This model indicates that EMT is two-step program, with two distinct phases: initiation and maintenance, controlled by different miRNA/transcription factor loops. Therefore, partial EMT is defined as initiated EMT without stabilization, with activated MiR-34/SNAIL loop and without activated MiR-200/ZEB loop, resulting in low miR-34 and ZEB; high SNAIL and miR-200 phenotype [55]. A second model proposes that MiR-34/SNAIL loop is an EMT/MET controller, that prevents aberrant EMT or MET initiation, however, it is not triggering any phenotypic changes itself. In this model, the MiR-200/ZEB loop with the input of SNAIL behaves as a tri-stable or three-way switch allowing for the existence of three phenotypes: epithelial (high MiR-200, low ZEB), mesenchymal (low MiR-200, high ZEB) and hybrid E/M (medium MiR-200, medium ZEB) [56]. Experiments focused on full EMT/MET or epithelial/mesenchymal phenotypes are supporting both models, as both models provide similar characteristic for the epithelial and mesenchymal phenotypes [51,52,57]. There is still a need for research focused on hybrid E/M phenotype and mechanisms of its stabilization. Despite unclear mechanism of hybrid E/M phenotype acquisition, it is known that on some point cancer cells co-express both epithelial and mesenchymal markers [58]. Recently, it was recognized that cells which undergo such “uncoupled” EMT are associated with a higher metastatic risk than those after full EMT [59].

### 2.5. EMT Status in CTCs and Its Prognostic Value 

Heterogeneity in epithelial and mesenchymal markers among breast cancer CTCs was reported in many studies [58,60,61,62,63,64,65]. Moreover, it was reported, that the distribution of these markers in CTCs may be changing during disease progression and treatment [65]. Not surprisingly, CTCs found in the blood of patients with different breast cancer subtypes have diverse phenotypes. For patients with luminal cancers CTCs were predominantly epithelial, while for other subtypes predominantly mesenchymal [58]. Despite many studies, it is unclear which of those two distinct phenotypes may be prognostic for treatment response, PFS and OS. Some research show that mesenchymal phenotype is associated with poor outcome and shorter PFS [58,65,66,67], but other studies suggest that prognostic value is rather associated with the epithelial phenotype [40,62]. Interestingly, the presence of mesenchymal CTCs was described a predictive factor, regardless of breast cancer subtype [58].

It seems that the hybrid phenotype may be the most predictive. Co-expression of E and M markers in CTCs is associated with cancer progression, metastasis and shorter PFS [60,61,62,63]. The E/M population was enriched post-chemotherapy, which corresponded with the lack of response to treatment. The detection of this CTCs subpopulation was an independent predictive factor for reduced PFS, whereas in the HER2-negative cohort, it was also predictive for decreased OS [63]. 

### 2.6. CTC Clusters

It was demonstrated in mouse models that breast cancer metastases are of polyclonal origin [68,69], which is inconsistent with the dissemination and expansion of a single cell, but can be easily explained by the dissemination of CTC clusters. CTC clusters are very rare (about 10 times more rare than single CTCs), but up to 50 times more metastatic than single CTCs [68]. Molecular profiling of single and clustered CTCs revealed high expression of desmosomal junction protein plakoglobin in clusters [68], which may be due to its function in maintaining strong cell–cell contacts. In another study, metastases were shown to arise from tumor cells’ clusters highly expressing keratin14. Interestingly, transcriptional program of K14+ cells also leads to the enrichment in desmosomal fraction [69]. Despite high expression of some epithelial traits, CTC clusters express also mesenchymal markers, which suggest a shift towards the hybrid phenotype [58,70]. Additionally, Gkountela et al., have demonstrated that CTCs clustering leads to hypomethylation of binding sites typical for master stemness and proliferation regulators (OCT4, NANOG, SOX2, and SIN3A)—leading to their enhanced expression—and hypermethylation of Polycomb target genes [71].

The origin of CTC clusters is a subject of a debate. Originally, it was surmised that CTC clusters arise by collective invasion and dissemination of oligoclonal groupings of tumor cells [68,69], but recent findings revealed that they can be formed by tumor cells aggregation within vasculature. This aggregation is induced by homophilic interactions of CD44 molecule, the receptor for hyaluronic acid [72].

CTC clusters contain not only tumor cells, but also cancer-associated fibroblasts (CAFs), white blood cells and platelets [73]. This heterogeneity may serve many purposes: help to eliminate anoikis by providing cell–cell contacts ([73,74], physically shield from shear stress and immune attack, promote the transition to hybrid phenotype by paracrine stimulation of tumor cells with TGF-β [75] and promote efficient cell seeding in the secondary site by providing its own microenvironment [76].

The question of how relatively big CTC clusters may traverse capillaries and be efficient in getting into secondary site was resolved by the work of Au et al, [77] who demonstrated the ability of the clusters to re-organize and form a single-file chain, which, after extravasation, may re-organize again.

Interestingly, the treatment with Na^+^/K^+^-ATPase inhibitors (ouabain and digitoxin, cardiac glycosides) resulted in cluster dispersion and led to the suppression of spontaneous metastasis formation in mice, which may represent a future treatment approach [71].

### 2.7. Early Breast Cancer; Prognostic Value, Programs and Trials

The clinical value of CTCs in an early setting is limited by their scarcity. The small number of CTCs per ml of peripheral blood is the major challenging physical limit. CTCs are particularly rare in non-metastatic breast cancer, usually less than 1 CTC/10 mL of blood is found [78], with five or more CTCs being a rare event (1–5.9%) [79]. CTCs are detectable in about 20% to 25% of patients with localized nonmetastatic breast cancer at the time of diagnosis using a lower threshold (≥1 CTC per 7.5 mL blood) than in MBC (≥ 5 CTC per 7.5 mL blood) [79,80,81,82]. Similarly to MBC, CTCs provide independent prognostic information whether obtained before or after surgery, including after neoadjuvant or adjuvant chemotherapy.

In early breast cancer (EBC) the accumulated data about the biology of CTC release by the primary tumor are scarce. Neoadjuvant and adjuvant studies reported a moderate association of CTC detection with positive lymph nodes, but not with any of the other classical prognostic factors, nor with tumor subtype [79,82]. Tumor cell dissemination has been suggested to occur early in BC progression, even before the tumor has become invasive. A single study on 73 patients with either ductal or lobular carcinoma in situ reported that three patients (4.1%) had 1 CTC per 22.5 mL of blood [83].

The recent international meta-analysis (IMENEO study) based on more than 2000 nonmetastatic BC patients from 16 centers treated by neoadjuvant chemotherapy (NCT) aimed to assess the clinical validity of CTCs detection as a prognostic marker. CTC counts were found to be a strong and independent prognostic indicator for distant-metastasis-free survival, OS and locoregional relapses [79]. Importantly, the impact on survival was linked to the number of detected CTCs. The study revealed that statistical significance of CTC count was rising with CTC number, from none for one cell to HR of 6.25 (95% confidence interval (CI) = 4.34 to 9.09) for five or more cells [79], reinforcing the concept of CTC counts as a quantitative marker.

Prognostic significance of CTCs was also clearly demonstrated in adjuvant setting [78]. The randomized trial SUCCESS-A on more than 2000 patients revealed that CTC positivity before as well as after adjuvant chemotherapy was an independent prognostic factor, with poor disease-free survival (DFS) (hazard ratio (HR) = 2.28, 95% CI = 1.48 to 3.50) and OS (HR = 3.95, 95% CI = 2.13 to 7.32). Patients with at least 5 CTC per 30 mL showed the worst prognosis [81]. Similarly, pooled retrospective analysis of individual data from 3173 patients with stage I to III demonstrated that the presence of CTCs (> 1 per 7.5 mL blood) was an independent predictor of poor DFS (HR = 1.82, 95% CI = 1.47 to 2.26), distant DFS (HR = 1.89, 95% CI = 1.49 to 2.40), breast cancer-specific survival (HR = 2.04, 95% CI = 1.52 to 2.75), and OS (HR = 1.97, 95% CI = 1.51 to 2.59) [82]. 

SUCCESS-A and the ECOG-ACRIN study E5103 showed an increased risk of recurrence for patients with persisting CTCs two years and even five years after (neo)adjuvant chemotherapy [84,85]. The CTC-positivity after five years was the strongest predictor of late disease recurrence in patients with HR+ breast cancer that had no signs of disease recurrence, as presented by Sparano et al. in multivariate analysis [84]. Similarly, in SUCCESS-A study CTCs detected at 2 years (in 18.2% of patients) were associated with a 3.9-fold increased risk of death and a 2.3-fold higher recurrence risk in multivariable models [86]. Analysis of 206 patients enrolled in the same study with known CTC status at 5 years, revealed that CTCs-positivity was found in 7.8% and was associated with a 6-fold increase in recurrence [85].

Adding to the prognostic impact of CTC enumeration, a recent retrospective analysis of the SUCCESS-A trial and of the Surveillance, Epidemiology, and End Results database suggested thatCTC-positive patients may benefit from adjuvant radiation therapy in terms of relapse-free survival and/or OS, [87].

The clinical utility of CTC detection in early breast cancer patients was also investigated in a prospective trial, TREAT-CTC (NCT01548677). Up to date, this is the only study in the (neo)adjuvant setting testing the impact of CTCs detection results on treatment decisions. This trial aimed to test the impact of additional treatment (targeted therapy) for the elimination of persistent CTCs. Sixty-three patients with non-metastatic HER2-negative breast cancer and detectable CTCs (>1/7.5 mL blood) after neoadjuvant chemotherapy and surgery were randomized to trastuzumab (Herceptin) or observation. There was no difference in CTCs at week 18, the trial’s primary endpoint, and the trial was halted for futility [88]. This negative result might be related to the HER2 status of CTCs, since CTCs were not required to be HER2-positive for inclusion in the study. In fact, the majority of the patients (76%) had HER2-negative CTCs. Results obtained might suggest that the failure of the Treat CTC trial was rather related to choosing an inappropriate treatment intervention than reflecting a general failure of the concept of CTC-based treatment decisions.

### 2.8. Advanced Breast Cancer; Prognostic Value, Programs and Trials

Prognostic value of CTCs has been demonstrated in metastatic breast cancer by many studies [18,89,90].

Since the Food and Drug Administration approval of CellSearch System (Menarini Silicon Biosystems, Inc, Bologna, Italy) as the only device suitable for CTC quantification in the clinics, many efforts were undertaken to asses CTC utility in the monitoring and treatment of women with MBC. According to Budd et al. estimation of CTC has several advantages over imaging methods in monitoring MBC [91]. In 2004 Cristofanilli et al. demonstrated that detection of CTCs has prognostic value during the course of the disease [92]. CTCs were detectable in approximately 60% of MBC patients, and a CTC count ≥5 cells per 7.5 mL of blood was associated with significantly worse progression-free survival (PFS) and OS, providing evidence for its clinical validity. This finding was an impulse to conduct several translational research projects investigating the role of CTC in response to therapy. Currently, numerous clinical trials are addressing this issue, most (41%) in the USA, 12% in China, 10% in Germany and 9% in France [93]. In some of the trials, CTC are only enumerated [94,95] whereas other also asses their phenotype [96,97]. They are attempting to answer the question if either CTC number or/and CTC phenotype may serve as a criterion for therapy decisions. 

SWOG S0500, a prospective clinical trial, addressed a question whether patients with persistently high CTC number (≥ 5 cells/ 7.5 mL of blood) after one cycle of first-line chemotherapy will benefit of switching to an alternate therapy. The trial failed to demonstrate improved survival after early change of chemotherapy relative to continuation of the same therapy [95]. Thus, persisting CTCs might represent chemoresistant tumor cells which require alternative methods of elimination. Recently, the prospective-retrospective study was conducted of the SWOG trial to test whether a presence of CTC clusters is associated with poor prognosis [98]. A conclusion of this study is that the presence of CTC clusters has no effect on outcomes of MBC patients starting first-line chemotherapy, raising the question as to whether clusters mediate the metastatic process as has been postulated by other researchers [66,68]. These data rather suggest that more crucial in progression of the disease is the total number of CTC.

MBC is a heterogeneous disease therefore stratification of MBC patients for treatment decisions and novel therapies development is of most importance. In a retrospective study analysing data from 18 international centres, including 2436 MBC patients, Cristofanilli et al. demonstrated that CTC are ideal for stratification of patients in stage IV of the disease. Stage IV indolent patients (< 5CTC) had longer OS than those in stage IV aggressive (> 5CTC, 36.3 months vs. 16 months) independent of disease subtype [99]. Thus, CTC enumeration may serve as important tool for staging of advanced disease and for disease stratification.

The second completed trial is a STI-CTC III phase study from French group [94]. This study was set up as a strategy trial to test whether CTC count could help customize the choice between first-line hormone therapy (HT) or chemotherapy (CT) in ER+ HER2- MBC patients. Patients were randomized between clinically-driven choice in the standard arm (no CTC count, physician’s choice of HT or CT based on clinical factors), or a CTC-driven treatment arm (HT if < 5 CTC/7.5mL or CT if ≥ 5 CTC/7.5mL). Patients who received CT based on CTC count had longer PFS than those whose treatment was clinically-driven. The median PFS was 10.5 months with HT in the clinically- driven arm for those with a high CTC count versus 15.5 months with CT in the CTC-driven arm and showed a trend toward longer OS (median, 37.1 vs. 42.0 months, respectively). Although the results of this trial are promising and show that including of CTC in the treatment decision might improve patients’ outcome in particular cases, this study has some limitations. It was conducted prior to the implementation of new endocrine treatment with CDK4/6 inhibitors and it lacks standardized clinical criteria for CT in the clinically- driven arm.

Other randomized trials are still ongoing, as reviewed elsewhere [100].

The CirCe01 study is another trial evaluating the response to chemotherapy based on CTC number after the first cycle. This multicenter phase III study included MBC patients who were CTC-positive (≥5 CTCs/7.5 mL) after two lines of chemotherapy. Patients were randomized into two arms: either followed by the clinical assessment or by the determination of CTCs. Those in the CTC-driven arm were switched to another chemotherapy if CTC count did not decrease after one cycle of chemotherapy [101] to save them from ineffective and toxic treatment.

The DETECT study is addressing the issue of significance of CTC phenotype for treatment decisions in MBC. This study provides a prospective, multicenter, clinical trial program comprising two phase III studies (DETECT III and V) and one phase II study (DETECT IV) [97]. In DETECT IVa, postmenopausal women with HR-positive MBC are treated with everolimus and an endocrine therapy, while in DETECT IVb women with triple-negative MBC or HR-positive tumors with an indication for chemotherapy receive eribulin. In DETECT V/CHEVENDO interventions are not based on CTCs presence [100]. This translational project aims to estimate an endocrine-responsiveness-score (ERS) based on ER and HER2 expression in CTCs to create a tool for predicting treatment outcome in the HR-positive, HER2-positive disease. Patients recruited to this trial receive dual anti-HER2 blockade in combination with either chemotherapy or endocrine therapy plus CDK4/6 inhibitor ribociclib [93].

In conclusion, although the detection and characterization of CTC give promise to influencing treatment decisions in MBC, it is too early to announce success in this field. The ongoing trials should provide much more data to evaluate CTC clinical utility.

All clinical trials assessing the impact of CTC-based treatment decisions described in this review are summarized in Table 1.

### 2.9. Current Technical Limitations in CTC Clinical Research

The EpCAM-dependent CTCs detection and isolation system CellSearch is the most popular and the only FDA approved platform for CTC clinical usage. Its detection rate and prognostic relevance in breast cancer is quite satisfactory, but there is still a debate about its incapability of detection of CTCs which have lost epithelial markers (a common occurrence in some other cancers, for example in renal cell carcinoma) or its incompatibility with tumors of non-epithelial origin. In breast cancer EpCAM^low^, CTCs seem to be not significant for survival, but EpCAM-independent approaches (comprehensively reviewed by Gabriel et al., [102]) achieve higher detection rates. For the CellSearch platform the CTC detection rate is about 20–30% in EBC to 60–70% in MBC [93]. Similar rates are obtained for filtration-based methods [103,104]. With leukapheresis, these numbers could be increased to as high as 90% for EBC [105]. In label-free multifluidic platform the numbers are, respectively, about 74% for EBC and 81% for MBC [106]. On the other hand, in another EpCAM-dependent method, Adnatest (immunoenrichment and epithelial mRNA detection), the detection rate was about 40% for MBC and the assigned prognostic value for these CTCs was low to none [107]. Some other EpCAM-independent approaches like CytoTrack await large cohort studies. Overall, so far there is no hard evidence for the prognostic advantage of the EpCAM-independent approaches in BC. Interestingly, there might be another way to bypass the problems with EpCAM-dependent isolation: finding some other, more universally expressed marker, like, for example the malaria protein VAR2CSA [108].

## 3. Conclusions

Although detection of CTCs is a rare event in nonmetastatic breast cancer, its clinical validity as a prognostic marker has been repeatedly confirmed and remains unquestionable. The clinical utility of CTC detection still has not been reached. There is currently insufficient data to support the use of CTC status as a criterion for therapy decisions and it still remains to be investigated in prospective trials that show how it can be used for a prediction of therapy response and an improved clinical outcome. However, the results obtained so far in completed trials allow to anticipate that it can be included in clinical practice in the foreseeable future, especially in a form of monitoring non-metastatic patients after initial intervention (Figure 3). Furthermore, the identification of potential targets for more individualized treatment options might improve the use of CTCs in clinical practice.

## Figures and Tables

**Figure 1 ijms-21-01671-f001:**
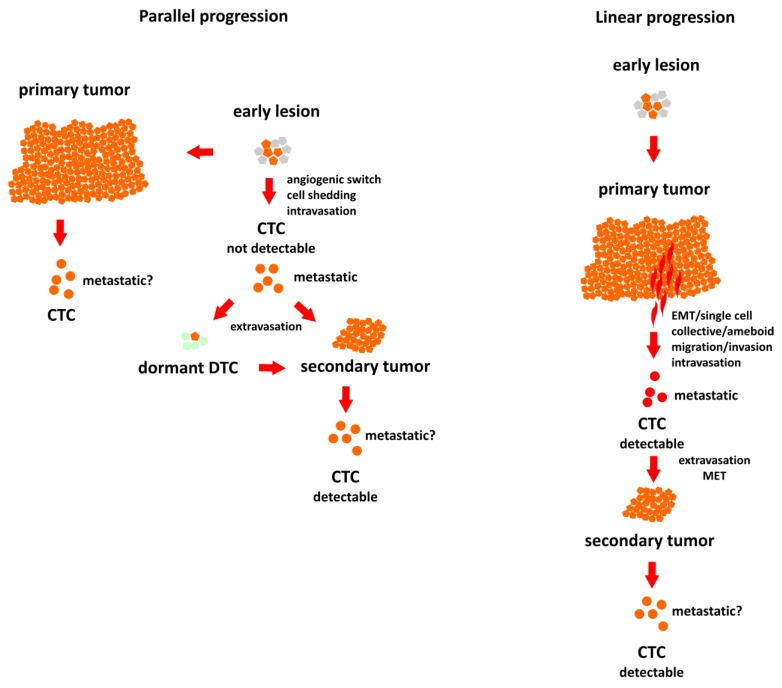
Two different models of breast cancer progression. In the parallel progression model dissemination occurs early and the metastatic circulating tumor cells (CTCs) are impossible to detect with the current sensitivity of the methods. In linear progression model metastatic CTCs are detectable before actual progression occurs.

**Figure 2 ijms-21-01671-f002:**
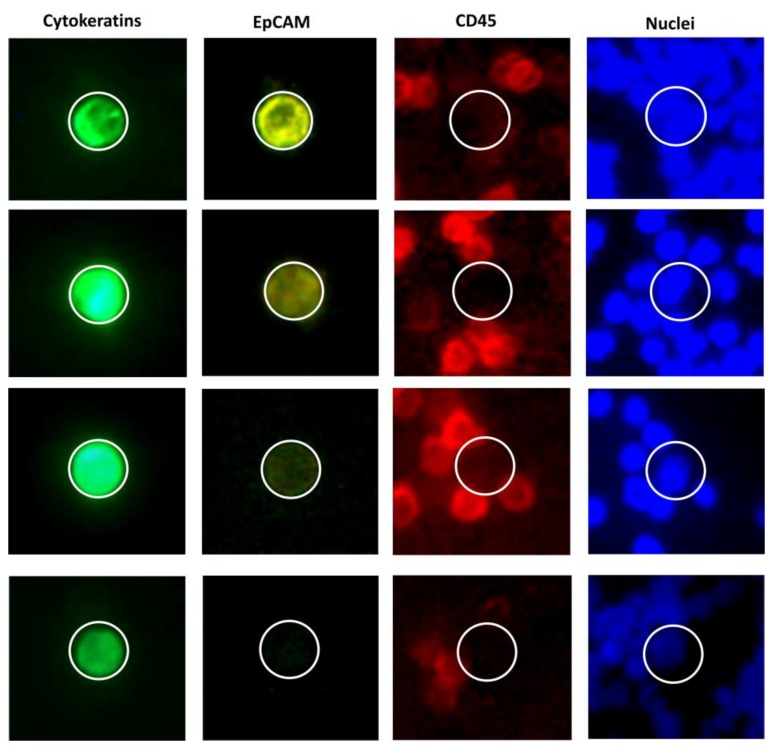
Heterogeneity of EpCAM expression in CTCs in metastatic breast cancer (luminal). Every set of images (Cytokeratins [pan-CK, EpCAM, CD45, nuclei [DAPI]) represents a CTC derived from a different patient. Most of the cytokeratin-positive cells have at least slight microscopically-detectable EpCAM expression, but there is also a population of EpCAM-negative CTCs. Images taken using EpCAM-independent CytoTrack system (unpublished, Malgorzata Szostakowska-Rodzos).

**Figure 3 ijms-21-01671-f003:**
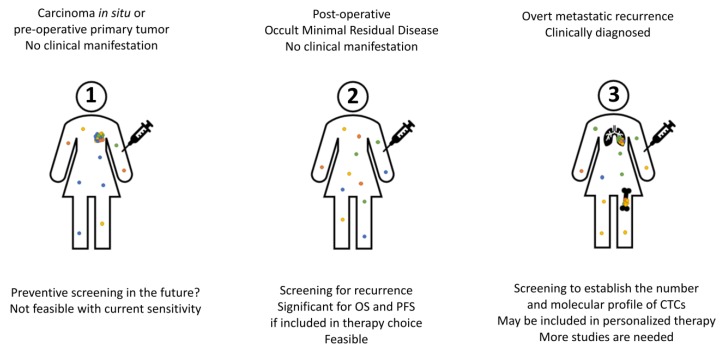
Possible use of CTC assessment in clinical practice. (**1**) Screening the healthy population may represent a future approach for early detection, but more sensitive methods are needed for this approach to be feasible. (**2**) Monitoring post-operative patients to flag possible recurrence is feasible and currently tested in clinical trials. (**3**) At present in metastatic breast cancer CTC analysis may be used for patients’ stratification. Molecular characterization of tumor cells may be used in the future to adjust the treatment.

**Table 1 ijms-21-01671-t001:** Clinical trials investigating CTC role in treatment decision in early and advanced breast cancer described in this review.

Breast Cancer Stage	Name	Trial Number	Reference
Early	TREAT-CTC	NCT01548677	[88]
Advanced	SWOG S0500	NCT00382018	[95,98]
	CirCe01	NCT01349842	[96]
	STIC-CTC	NCT01710605	[94]
	DETECT III	NCT01619111	[97]
	DETECT IV	NCT022035813	
	DETECT V	NCT0234447	

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
