# Peer review of "Circulating Tumor Cells in Early and Advanced Breast Cancer; Biology and Prognostic Value"

_ijms, 2020, doi:10.3390/ijms21051671_

Round 1

Reviewer 1 Report

It is a very educational article that explain  the pathogenesis of CTC and the change in mesenchymal epithelium that induces the spread and metastatic capacity of the tumor.

Figure 1 is very didactic but in Figure 2 I think it is necessary  more explanation of the events in the different photos.

Finally, I also think it would be interesting to have a table that summarizes the different studies with advanced breast cancer and CTC.

Author Response

Reviewer  1:

It is a very educational article that explain  the pathogenesis of CTC and the change in mesenchymal epithelium that induces the spread and metastatic capacity of the tumor. 

Figure 1 is very didactic but in Figure 2 I think it is necessary  more explanation of the events in the different photos.

Finally, I also think it would be interesting to have a table that summarizes the different studies with advanced breast cancer and CTC.

Response:

  1. The legend of Figure 2 was expanded to provide more explanation on a described heterogeneity.
  2. A table was added, but it encompasses not only advanced but also early BC. By separating these trials it underlines the differences between these two types of studies, and describing these differences is one of the main topics of this review. The table contains relatively little information, but we did not wish to replicate a very informative and comprehensive table published by Schochter et al., (2019) which probably cannot be improved, since no trials were started since then.

Reviewer 2 Report

This is a comprehensive review for current CTC research and clinical applications in breast cancer.  To make the review better, I suggest that the authors comment more on the current technical problems in CTC detection and isolation.  For example, the authors pointed out that not all CTC express high level of EpCAM due to the EMT status of the CTCs.  Yet, most of the CTC studies still rely on EpCAM.  Alternative CTC isolation/detection strategy should be discussed and their CTC detection efficiency should be compared with the EpCAM-method in prediction/prognosis correlation.  

Author Response

Reviewer 2

This is a comprehensive review for current CTC research and clinical applications in breast cancer.  To make the review better, I suggest that the authors comment more on the current technical problems in CTC detection and isolation.  For example, the authors pointed out that not all CTC express high level of EpCAM due to the EMT status of the CTCs.  Yet, most of the CTC studies still rely on EpCAM.  Alternative CTC isolation/detection strategy should be discussed and their CTC detection efficiency should be compared with the EpCAM-method in prediction/prognosis correlation.  

To make the review better we have added a paragraph pertaining to technical problems with CTC detection (focusing on breast cancer), including detection rates with different methods and how some of the problems can be addressed. However, we did not want to expand this paragraph further, since there are many reviews analyzing technical side of CTC research in great detail and we could not compete with that.